# The Mediating Effect of Community Identity between Socioeconomic Status and Sense of Gain in Chinese Adults

**DOI:** 10.3390/ijerph17051553

**Published:** 2020-02-28

**Authors:** Yanli Wang, Chao Yang, Xiaoyong Hu, Hong Chen

**Affiliations:** 1Faculty of Psychology, Southwest University, Chongqing 400715, China; wyl1054616437@163.com (Y.W.); yangchaopsy632@163.com (C.Y.); huxiaoyong@swu.edu.cn (X.H.); 2School of Psychology, Guizhou Normal University, Guiyang 550025, China; 3Key Laboratory of Cognition and Personality (Ministry of Education), Southwest University, Chongqing 400715, China; 4Research Center for Psychology and Social Development, Southwest University, Chongqing 400715, China

**Keywords:** socioeconomic status, community identity, sense of gain, social identity approach, social cure effect

## Abstract

Background: Several studies have explored the positive relationship between socioeconomic status and sense of gain. However, little is known about the underlying mechanism between them. This study aimed to explore whether community identity had a mediating role between them among Chinese adults. Methods: Data were collected from a nationally representative samples of 28,300 adults from the China Family Panel Studies. Socioeconomic status was assessed using individuals’ income and social status. Community identity was assessed through evaluation of the community’s public facilities, surrounding environment, surrounding security, neighborhood relationship, neighborhood assistance and feelings towards the community. Sense of gain was measured by evaluation of environmental conservation, gap between the rich and the poor, employment, education, medical treatment, housing, social security, and government corruption. Pearson’s correlation was used to examine the associations between major variables. Mediation analyses were performed to explore the mediating role of community identity between socioeconomic status and sense of gain. Results: Socioeconomic status was positively associated with sense of gain. Community identity played a mediating role between socioeconomic status and sense of gain. Conclusion: Community identity mediated the relationship between socioeconomic status and sense of gain. Promoting the mobility of socioeconomic status and actively intervening in community identity are conducive to improve sense of gain.

## 1. Introduction

Sense of gain, as a Chinese concept proposed in recent years, has attracted more and more researchers’ attention. Sense of gain refers to the improvement of material living standards brought by national development, including not only housing, education, medical care and social security, but also the right to enjoy fairness and justice [1]. Some studies [2] also believe that sense of gain is an overall feelings about the benefits of economic and social development, the realization of social equity and justice, and the supply of basic public services. Wen and Liu [3] measured sense of gain throughgap between the rich and the poor, government corruption and basic public services such as education, pension, medical treatment, housing, employment and social security. Other researchers [4] considered that sense of gain should be composed of economic sense of gain, public service sense of gain, political sense of gain, security sense of gain and self-realization sense of gain. Studies have reported there is positive associations between sense of gain and cognition and psychological health, such as well-being [5,6], government trust [2,7], andperception of social stability [8]. Therefore, exploring how to enhance individuals’ sense of gain contributes to national stability and psychological health.

Socioeconomic status is a social classification used to reflect the relative position of an individual in the social hierarchy, which is composed of objective material resources (often measured by income, education level, and occupational status), and subjectively perceived social status [9,10]. Socioeconomic status affects every aspects of an individual’s life, including sense of gain. For example, Wang [11] found that the increase of socioeconomic status was often accompanied by the improvement of sense of gain. Specifically, upper-middle status or upper- status residents have higher sense of gain than lower status residents. Wang, Tan, and Fu [12] also found that socioeconomic status had a significant and positive effect on sense of gain. Tan, Wang and Zhang [13] found significant differences in sense of gain among different educational levels and monthly income. Dou, Dong and Tan [14] found that there were positive relationship between subjective and objective socioeconomic status and sense of gain. There are also studies that provide indirect evidence for the relationship between socioeconomic status and sense of gain. Empirical research has confirmed that people with lower socioeconomic status had more negative affection and psychological symptoms (such as depressive symptoms, anxiety) [15,16,17,18,19] and lowerwell-being [20,21]. For example, a longitudinal study found that lower socioeconomic status was associated with higher rates of depressive symptoms, and that among individuals with depressive symptoms, those with lower socioeconomic status were more likely to worsen and last longer [22]. Moreover, individuals with lower socioeconomic status have lower satisfaction with life [20]. Other researcher [21] also found women with lower wealth status reported poor quality of life and happiness.

Despite mounting evidence that socioeconomic status is positively related to sense of gain, few studies to date have explored the potential mediating effect between socioeconomic status and sense of gain. Social identity approach, including social identity theory [23] and self-categorization theory [24], pointted out the mechanism of social group influence on individuals’ psychology. For instance, using longitudinal two-wave design and field-experimental design, researcher [25] found social identity positively affect perceived social support, which, in turn, positively affect collective self-efficacy, thereby reducing individuals’ ill-health outcomes, such as emotional exhaustion, chronic stress, and depressive symptoms. An emerging research approach, named as the social cure effect, claimed that social identity would promote well-being and mental health, alleviate depression [26,27,28,29,30]. Moreover, the social cure effect can be found in diverse group and context, such that social identity was associated with depressive symptom among older adults [31], greater satisfaction with life among homeless people [32] and more hedonic and eudaimonic well-being among voluntary and forced refugees [33]. Specific to the particular group, the social cure effect have also been found [34,35,36]. For example, used data from the six wave of the world values survey (2014), Greenaway et al. [29] concluded community identity could facilitate perceived personal control, these relationships can significantly improve well-being. Kearns et al. [37] examined 626 samples of urban centers in Ireland and demonstrated that subjective identity with community and religious groups were positively related to perceived social support and consequently, lower perceived stigma of mental ill-health. In conclusion, group identity can positively affect individual mental health and well-being. Moreover, there is a certain relationship between sense of gain and well-being. Specifically, sense of gain is embedded in well-being and is an important basic component about it [38]. Then, we can infer that community identity can also positively predict residents’ sense of gain.

In sum, although a number of studies have suggested that there is positive relationship between socioeconomic status and sense of gain, to the best of our knowledge, no studies have yet to explore the underlying mechanism between them. Based on social identity approach, this study aimed to establish an integrated theoretical framework that explained the relationship among socioeconomic status, community identity, and sense of gain, so as to bridge the knowledge gaps in existing literature. Specifically, we examined two major research questions. First, we examined whether socioeconomic status can directly affect sense of gain of Chinese adults. Second, we tested the underlying mechanism between socioeconomic status and sense of gain. We hypothesized that socioeconomic status was positively associated with sense of gain, and that this relationship was mediated by community identity (Figure 1).

## 2. Materials and Methods

### 2.1. Data and Study Population

The data are from China Family Panel Studies (CFPS), funded by 985 Program of Peking University and carried out by the Institute of Social Science Survey of Peking University. which is a nationwide, large-scale and multidisciplinary social tracking survey project, involving 25 provinces/cities/autonomous regions, with a target sample size of 16,000 households. The respondents include all family members in the sample households. CFPS questionnaire consists of community questionnaire, family questionnaire, adult questionnaire and children questionnaire. Adult questionnaires were used in this study. A total of 36,892 adults were surveyed in CFPS (2016), and 28,300 valid questionnaires were finally collected, excluding incomplete answers, refusal to answer, ignorance and invalid questionnaires of students. Age ranged from 18 to 99 years old (M = 46.50 years, SD = 15.75), 14,566 (51.50%) were males. Full descriptive statistics of the samples were presented in Table 1.

### 2.2. Measures

#### 2.2.1. Socioeconomic Status

In 2016 CFPS, socioeconomic status was measured by the items “What is your personal income here?” and “What is your social status here?” Participants were asked to rate their actual situation on a scale of 1 to 5 (1 = very low, 5 = very high). With reference to Tan and Kraus [39], the two indicators were standardized into standard scores, and then added. Higher scores indicated higher level of socioeconomic status.

#### 2.2.2. Community Identity

Community identity was measured by asking the questions in the 2016 CFPS: “what is the overall situation(such as neighborhood relationship, surrounding security, surrounding environment, and community facilities including eduation, medical treatment and transportation) around your community?”. The options are divided into five dimensions, including 1 = very good, 2 = good, 3 = neutral, 4 = worse, 5 = very bad. Neighborhood assistance was measured by asking the question. “If you need help from your neighbors, do you think they will?” with responses of “1 = must have, 2 = possible, 3 = not sure, 4 = possible not, 5 = most not”. Feeling towards the community was measured by asking the question. “Do you have feelings for your community” with responses of “1 = very emotional, 2 = more emotional, 3 = neutral, 4 = less emotional, 5 = very unemotional”. We arranged all the scores to be in the same direction, and higher scores indicated higher community identity. The Cronbach’s alpha of the scale in this study was 0.68.

#### 2.2.3. Sense of Gain

With reference to existing studies [3,4], sense of gain was measured by asking the questions: “How serious do you think the following problems are in our country” in this study, the following questions include environmental conservation, gap between the rich and the poor, employment, education, medical treatment, housing, social security, and government corruption. Each of the items is accompanied with eleven options, and the value of each option is from 0 (not serious) to 10 (very serious). In order to better reflect the meaning of this concept, we conducted the reverse scoring, and higher scores indicated higher sense of gain. The Cronbach’s alpha of the scale in this study was 0.84.

#### 2.2.4. Sociodemographic Factors

Several sociodemographic factors were taken into account, including gender (0 = male, 1 = female), age, household register (0 = agriculture account, 1 = non-agricultural account), Urban and rural classification (0 = rural, 1 = urban), and marital status (0 = never married, 1 = have a spouse (in marriage), 2 = cohabit, 3 = divorced, 4 = widowed).

### 2.3. Statistical Analyses

We used SPSS software version 22 (IBM, Armonk, NY, USA) to manage and analyze data. First, Harman’s single-factor test was used to examine whether there was the common method bias. Then, one-way analysis of variance (ANOVA) was conducted to explore the relationship of categorical variables with sense of gain. Third, descriptive statistics and correlation analysis was conducted to investigate the relationship among socioeconomic status, community identity, and sense of gain. Finally, mediation analyses were performed using a bootstrapping approach with the SPSS macro PROCESS based on 5000 bootstrap sample and a 95% confidence interval [40]. A *p* value <0.05 was considered significantly.

## 3. Results

Due to the use of self-reported data, Harman’s single-factor test was used to rate common method bias. Exploratory factor analysis revealed the first factor accounted of 21.15% of the total variance and did not explain most of the variance (<40%). Results showed that there was no common method bias in this study.

### 3.1. Preliminary Analyses

ANOVA was conducted to explore whether sense of gain differed between male and female, between agriculture account and non-agricultural account, between rural and urban, and between marital status. Results showed that there was no significant difference in sense of gain between male and female (*F* (1, 28,299) = 1.34, *p* > 0.05). Significant difference was found for household register, with agricultural people being significantly more sense of gain than non-agricultural people (*F* (1, 28,299) = 287.88, *p* < 0.01, and $\eta p^{2}$ = 0.01). Sense of gain was significantly different between rural and urban, namely, rural individuals had more sense of gain than urban individuals (*F* (1, 28,299) = 287.884, *p* < 0.01, and $\eta p^{2}$ = 0.01). The significant difference was found for marital status, with widowed being having significantly more sense of gain than others (*F* (4, 28,295) = 133.94, *p* < 0.01, and $\eta p^{2}$ = 0.02).

As indicated in Table 2, results showed that age (r = 0.30, *p* < 0.01) was positively associated with sense of gain. Socioeconomic status was positively associated with community identity (r = 0.20, *p* < 0.01) and sense of gain (r = 0.11, *p* < 0.01). And community identity was positively associated with sense of gain (r = 0.17, *p* < 0.01). 

### 3.2. Mediating Effects of Community Identity

To test the mediation effect, we conducted a mediation analysis using bootstrapping procedures proposed by Preacher and Hayes with the PROCESS macro in SPSS [40]. Table 3 and Figure 2 displayed the bootstrap results of the mediating effect of community identity between socioeconomic status and sense of gain.

As shown in Table 3 and Figure 2A, socioeconomic status was associated with sense of gain(c, *β* = 0.07, 95% [0.059, 0.073]). As shown in Table 3 and Figure 2B, the socioeconomic status-community identity path (a, *β* = 0.11, 95% [0.108, 0.121]) and the community identity-sense of gain path (b, *β* = 0.16, 95% [0.145, 0.168]) were significant. The socioeconomic status-sense of gain path (c’, *β* = 0.05, 95% [0.041, 0.055]) was also significant. The effect size of the variable was 0.01, 95% [0.005, 0.007]. In sum, results supported our hypothesis that community identity mediated the relationship between socioeconomic status and sense of gain.

## 4. Discussion

In this study, data from the 2016 CFPS are used to investigate whether and how socioeconomic status affects sense of gain and to determine whether community identity plays a mediating role between them. To our knowledge, this is the first national, large scale study to examine the potential mediating role of community identity on the relationship between socioeconomic status and sense of gain. Our results reveal that socioeconomic status is positively associated with sense of gain, and community identity plays a mediating role in the relationship between them.

An apparently unexpected finding in the study was that rural individuals had more sense of gain than urban individuals. In recent years, China has implemented the strategy of rural revitalization, with continuous economic development in rural areas and gradual improvement in the social security system. Sense of gain can also be the result of vertical social comparisons, where current conditions are compared to historical conditions. As a result, individuals in rural areas may have higher sense of gain than urban individuals [4,12]. And another unexpected finding was that widowed being had significantly more sense of gain than others. Widowed beings get more help from society. And because widowed being tend to be older, they are more likely to experience the positive effect of China’s rapid economic growth [12], which enhance their sense of gain. 

One of the main findings of this study is that socioeconomic status is positively associated with sense of gain. This result is supported by previous empirical studies, which reveal that socioeconomic status have a significant predictive effect on higher sense of gain [11,12,13,14], lower negative affection and psychological symptoms(such as depressive symptoms, anxiety) [15,16,17,18,19], and higher well-being [20,21]. Moreover, Qiu [41] pointed out that the important issue concerning socioeconomic status is, “who gets what? and how?” which to some extent, corresponds to sense of gain [14].

Another finding in this study is that socioeconomic status can indirectly influence sense of gain through the mediating role of community identity, which confirmed our hypothesis. Socioeconomic status is a social classification used to reflect the relative position of an individual in the social hierarchy. A person’s socioeconomic status reflects the objective social resources he/she posss and his/her subjective perception of his/her social status [9,10]. The perspective of social cognition of socioeconomic status points out that people with low socioeconomic status have less social resources, less educational opportunities, poor living environment, and face more uncertainties and unpredictability in life, such as unemployment [9]. In addition, people with low socioeconomic status perceive greater pressure in close relationship [42], lower trust level [43], increase adverse childhood experience/maltreatment [44], and face more community violence and higher conflicts [45], which are not conducive to the improvement of community identity for people with low socioeconomic status. And that the social cure effect suggest that low community identity can cause individuals’ ill-health outcome [25], enhance depression [26,29,30], decrease satisfaction with life [32] and well-being [30,33]. Therefore, community identity mediates the effect of socioeconomic status on sense of gain. Namely, a decreased socioeconomic status is associated with a reduced community identity, which then predicts relatively low sense of gain.

Our results have important implications for the development of theory and the formation of policies. This study revealed that community identity mediate the relationship between socioeconomic status and sense of gain. This study not only provides empirical evidence supporting the social cure effect but also enriches the underlying mechanism of socioeconomic status on sense of gain by building an integrated theoretical model. Moreover, our results reveal that socioeconomic status cannot directly affect sense of gain, but can indirectly influence sense of gain by the mediating effect of community identity. Promoting the mobility of socioeconomic status and actively intervening in community identity are conducive to improving sense of gain.

Overall, this study establishes an integrated framework to examine the mediating effect of community identity on the relationship between socioeconomic status and sense of gain. However, several limitations of this study must be noted. First, data from 2016 China Family Panel Studies is a second-hand data. This may lead to the items selected to measure the variables cannot fully reflect the meaning of the concept. Specific research tools for sense of gain should be developed in future studies. Second, this study is a cross-sectional study and cannot be used for causal inference. Therefore, future studies can use longitudinal studies or laboratory experiments to explore the mediating role of community identity between socioeconomic status and sense of gain. Third, although we find significant relationship between socioeconomic status, community identity and sense of gain, the effect size is very small. Future research should focus on other variables that affect sense of gain and the underlying mechanisms between them.

## 5. Conclusions

The current study examined the mediating role of community identity on the relationship between socioeconomic status and sense of gain in Chinese adults. Results show that there is a positive relationship between socioeconomic status and sense of gain. Community identity play a mediating role between socioeconomic status and sense of gain. Promoting the mobility of socioeconomic status and actively intervening in community identity are conducive to improve sense of gain.

## Figures and Tables

**Figure 1 ijerph-17-01553-f001:**
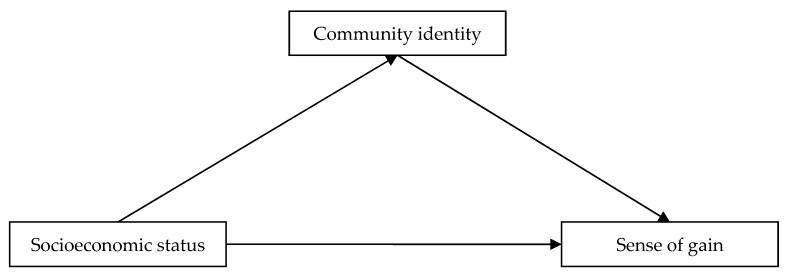
Basic model of the relationship between socioeconomic status and sense of gain as mediated by community identity.

**Figure 2 ijerph-17-01553-f002:**
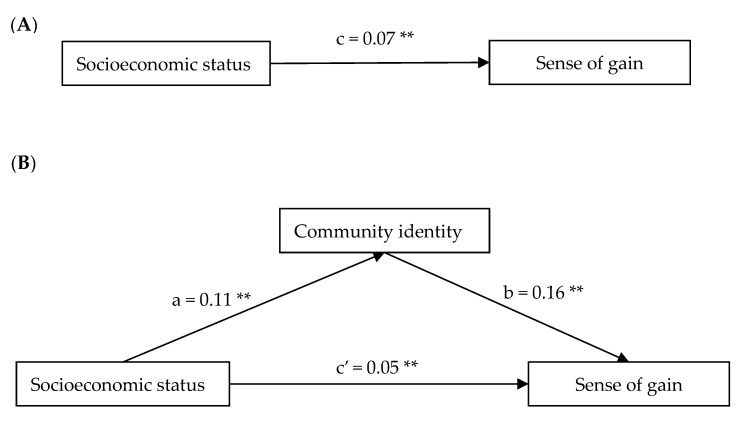
Tests of the theorized mediation model. (**A**) The total effect (socioeconomic status predicting sense of gain). (**B**) The indirect effect (with community identity as mediator). Indirect effect, a × b (β) = 0.02, *p* < 0.01. Standardized regression coefficients are displayed. * *p* < 0.05, ** *p* < 0.01.

**Table 1 ijerph-17-01553-t001:** Demographic characteristics of sample.

Variables	Overall Sample (*N* = 29,093)	
	*N*	%
Gender		
Male	14,566	51.5
Female	13,734	48.5
Household register		
Agriculture account	20,531	72.5
Non-agricultural account	7769	27.5
Urban and rural classification		
Rural	14,228	50.3
Urban	14,072	49.7
Marital status		
Never married	2768	9.8
Have a spouse (in marriage)	23,437	82.8
Cohabit	110	0.4
Divorced	549	1.9
Widowed	1436	5.1

**Table 2 ijerph-17-01553-t002:** Correlations among central study variables.

	M ± SD	1	2	3
1. Age	46.50 ± 15.75			
2. Socioeconomic status	0.00 ± 1.70	0.10 **		
3. Community identity	22.19 ± 3.22	0.15 **	0.20 **	
4. Sense of gain	30.33 ± 13.88	0.30 **	0.11 **	0.17 **

Notes: ** *p* < 0.01 (two-tailed); * *p* < 0.05 (two-tailed).

**Table 3 ijerph-17-01553-t003:** Test of mediation effects of community identity on the relationship of socioeconomic status to sense of gain: Bootstrap results.

Path/Effect	Standardized			
	β	Boot SE	Boot LLCI	Boot ULCI
C (total effect)	0.07	0.01	0.059	0.073
*a* Socioeconomic status→Community identity	0.11	0.01	0.108	0.121
*b* Community identity→Sense of gain	0.16	0.01	0.145	0.168
*c’* Socioeconomic status→Sense of gain	0.05	0.01	0.041	0.055
*A* × *b* (indirect effect)	0.02	0.01	0.016	0.020
R-sq_med	0.01	0.01	0.005	0.007

Note. Bias corrected and accelerated 95% CI, bootstrap resamples = 5000. The 95% CI for the standardized result was produced with the bias corrected and accelerated option in the bootstrap dialogue box.

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
