# Peer review of "The Mediating Effect of Community Identity between Socioeconomic Status and Sense of Gain in Chinese Adults"

_ijerph, 2020, doi:10.3390/ijerph17051553_

Round 1

Reviewer 1 Report

This manuscript examines the relationship between “sense of gain” using a large dataset from a panel study of Chinese adults. SES is defined as “social status” and community identity is a complex mix of variables from the dataset including evaluation of community, public facilities, security and neighborhood relationship and overall feelings toward the community. Sense of gain is measured as the gap between rich and pool along with a range of other variables from the data set including employment, medical information, social security, housing and government corruption. Findings are that SES has a negative association with sense of gain with the community identity variable as a partial mediator. The authors conclude that promoting mobility of SES and enhancing community identity can improve sense.

While the use of a large-scale dataset is a clear strength of this work, the definitions of the key variables lacks clarity in terms of theory, justification based on prior research and any psychometric validation that they combine to form a coherent measure. This is especially a concern for the measure of sense of gain defined as the cognitive and emotional experience produced by the promotion, affirmation and material reward through individual efforts. However, there appears to be a disconnect between the conceptual definition of sense of gain and the variable extracted from the dataset and the authors do not provide sufficient detail to explain how and why these items were selected to measure sense of gain.

There are other examples of assertions made from the data that require some additional explain and linkage to existing research. This includes the assertion that SES is an “identity imprint”. The authors make a brief reference to social identity through but to not provide explanation as to SES is a social identity. How do we know that this is the case across all SES levels? Is the identity income or some other variable that was not included in the large data set? The lack of detail here makes the connection between social identity theory and the variables included in the measure of community identity and SES unclear and unsupported by key tenants of this theory.

Research on subjective well-being could be very useful here to provide a more robust theoretical framework to justify the notion of SES as an identity imprint and the inclusion of variables such as government corruption, social security and housing as measures of sense of gain.

Author Response

Response to Reviewer 1 Comments

Dear Ms. Erica Wang and the reviewer,

Thank you for your constructive comments on our paper (ijerph-699193). We are very grateful to the editor and reviewers for the excellent level of detailed feedback offered to enable us to enhance the manuscript. We have carefully addressed the comments of the reviewers and highlighted(in red)the main changes made in the revised paper. Thank you for the opportunity to resubmit our manuscript for further consideration for publication in International Journal of Environmental Research and Public Health. All responses are made as follows.

Sincerely

Comments to the Author

This manuscript examines the relationship between “sense of gain” using a large dataset from a panel study of Chinese adults. SES is defined as “social status” and community identity is a complex mix of variables from the dataset including evaluation of community, public facilities, security and neighborhood relationship and overall feelings toward the community. Sense of gain is measured as the gap between rich and pool along with a range of other variables from the data set including employment, medical information, social security, housing and government corruption. Findings are that SES has a negative association with sense of gain with the community identity variable as a partial mediator. The authors conclude that promoting mobility of SES and enhancing community identity can improve sense.

Point 1: While the use of a large-scale dataset is a clear strength of this work, the definitions of the key variables lacks clarity in terms of theory, justification based on prior research and any psychometric validation that they combine to form a coherent measure. This is especially a concern for the measure of sense of gain defined as the cognitive and emotional experience produced by the promotion, affirmation and material reward through individual efforts. However, there appears to be a disconnect between the conceptual definition of sense of gain and the variable extracted from the dataset and the authors do not provide sufficient detail to explain how and why these items were selected to measure sense of gain.

Response 1: We appreciate these suggestions. First, we have added relevant sentences about the definitions of the key variables(from line 48 to line 52). Second, we have added justification based on prior research(from line 39 to line 43). Third, we also have added psychometric validation that the combine to form a coherent measure(from line 134 to line 136 and from line 143 to line 145). Fourth, we have provided sufficient detail to explain how and why these items were selected to measure sense of gain(from line 138 to line 141). Finally, we have revised the conceptual definition of sense of gain to better connect it with the variable extracted from the dataset(from line 34 to line 39).

“Socioeconomic status is a social classification used to reflect the relative position of an individual in the social hierarchy, which is composed of objective material resources (often measured by income, education level, and occupational status) and subjectively perceived social status[9, 10]. Socioeconomic status affects every aspects of an individual's life, including sense of gain.”

“Wen and Liu[3] measured sense of gain through gap between the rich and the poor, government corruption and basic public services such as education, pension, medical treatment, housing, employment and social security. Other researchers[4] considered that sense of gain should be composed of economic sense of gain, public service sense of gain, political sense of gain, security sense of gain and self-realization sense of gain.”

“We arranged all the scores to be in the same direction, and higher scores indicated higher community identity. The Cronbach’s alpha of the scale in this study was 0.68.”

“In order to better reflect the meaning of this concept, we conducted the reverse scoring, and higher scores indicated higher sense of gain. The Cronbach’s alpha of the scale in this study was 0.84.”

“With reference to existing studies[3, 4], sense of gain was measured by asking the questions “How serious do you think the following problems are in our country” in this study, the following questions include environmental conservation, gap between the rich and the poor, employment, education, medical treatment, housing, social security, and government corruption.”

“Sense of gain, as a Chinese concept proposed in recent years, has attracted more and more researchers' attention. Sense of gain refers to the improvement of material living standards brought by national development, including not only housing, education, medical care and social security, but also the right to enjoy fairness and justice[1]. Some studies[2] also believe that sense of gain is an overall feelings about the benefits of economic and social development, the realization of social equity and justice, and the supply of basic public services.”

References

  1. People's Daily Online. How to enhance the sense of happiness and security (three questions for people's livelihood for people's better life⑧). Available online: http://society.people.com.cn/n1/2017/1109/c1008-29635156.html (accessed on 9 November 2017)

  2. Li, P.; Bai, W. C. Research on the influence of people's sense of gain on government trust. Admin Tri. 2019, (4), 75–81, doi:10.16637/j.cnki.23-1360/d.2019.04.010.

  3. Wen, H.; Liu, Z. P. Time comparison of the sense of gain to China: Trends and disparities empirical analysis based on Chinese urban and rural social governance data. J Soc Sci. 2018, (3), 3–20, doi:10.13644/j.cnki.cn31-1112.2018.03.001.

  4. Yang, J. L.; Zhang, S. H. Analysis of the General Social Survey Data on the Chinese people's sense of fulfillment. J M Stud. 2019, 188(3), 102–112.

  5. Kraus, M. W.; Piff, P. K.; Mendoza-Denton, R.; Rheinschmidt, M. L.; Keltner, D. Social class, solipsism, and contextualism: How the rich are different from the poor. Psychol Rev. 2012, 119(3), 546–572, doi: 10.1037/a0028756.

  6. Kraus, M. W.; Tan, J. J.; Tannenbaum, M. B. The social ladder: A rank-based perspective on social class. Psychol Inq. 2013, 24(2), 81–96, doi:10.1080/1047840X.2013.778803.

Point 2: There are other examples of assertions made from the data that require some additional explain and linkage to existing research. This includes the assertion that SES is an “identity imprint”. The authors make a brief reference to social identity through but to not provide explanation as to SES is a social identity. How do we know that this is the case across all SES levels? Is the identity income or some other variable that was not included in the large data set? The lack of detail here makes the connection between social identity theory and the variables included in the measure of community identity and SES unclear and unsupported by key tenants of this theory.

Response 2: Thank you for your constructive advice for improving our study. First, we have modified the relevant expressions to make them more clear (from line 48 to line 52). Second, we removed sentences unrelated to the purpose of the study.

“Socioeconomic status is a social classification used to reflect the relative position of an individual in the social hierarchy, which is composed of objective material resources (often measured by income, education level, and occupational status) and subjectively perceived social status[9, 10]. Socioeconomic status affects every aspects of an individual's life, including psychological and physical health.”

References

  1. Kraus, M. W.; Piff, P. K.; Mendoza-Denton, R.; Rheinschmidt, M. L.; Keltner, D. Social class, solipsism, and contextualism: How the rich are different from the poor. Psychol Rev. 2012, 119(3), 546–572, doi: 10.1037/a0028756.

  2. Kraus, M. W.; Tan, J. J.; Tannenbaum, M. B. The social ladder: A rank-based perspective on social class. Psychol Inq. 2013, 24(2), 81–96, doi: 10.1080/1047840X.2013.778803.

Point 3: Research on subjective well-being could be very useful here to provide a more robust theoretical framework to justify the notion of SES as an identity imprint and the inclusion of variables such as government corruption, social security and housing as measures of sense of gain.

Response 3: Thank you for your constructive advice for improving our study. We have added research on subjective well-being(from line 64 to line 66).

“individuals with higher socioeconomic status have higher satisfaction with life[20]. Other researcher[21] also found women with lower wealth status reported poor quality of life and happiness”

References

  1. Xu, W.; Sun, H.; Zhu, B.; et al. Analysis of factors affecting the high subjective well-being of Chinese residents based on the 2014 China family panel study. Int. J. Environ. Res. Public Health. 2019, 16(14), 2566, doi: 10.3390/ijerph16142566

  2. He, Z.; Cheng, Z.; Bishwajit, G.; et al. Wealth inequality as a predictor of subjective health, happiness and life satisfaction among Nepalese women. Int. J. Environ. Res. Public Health. 2018, 15(12), 2836, doi:10.3390/ijerph15122836

Reviewer 2 Report

The authors present a national, large scale study to examine the potential mediating of community identity on the relationship between socioeconomic status and “sense of gain”. In my opinion, here we have the main problem of this research. As the authors will recognize in their limitations paragraph (line 219), the items that they used for the concept “sense of gain” do not reflect the meaning of the concept, so they should consider changing to another concept. For instance, it would be more accurate to speak about “National problems perceptions” than “sense of gain”.

In the Introduction (line 36) the definition they offer from the “spiritual level” of the concept “sense of gain” it is politically biased. They back on three references, which are far from being the most accepted. Again, I would recommend not using the concept “sense of gain”; but if the authors still want to mention it, do not include that definition of the spiritual level (they might consider other definitions included in the “Handbook of the Psychology of Religion and Spirituality”).

I believe authors do the right decision when they introduce some of the classical theories on social identity (line 55).

In the methods, I would recommend that the authors arrange all the scores to be in the same direction, and the higher scores always represent higher values (Socioeconomic status, Community identity, Problems perceptions…).
In addition, the authors should inform about the reliability (Cronbach´s Alfa scores) of their composed variables (Socioeconomic status, Community identity, Problems perceptions…).

In their results, there is an important error. We cannot use correlations for categorical variables (line 148), it is very complicated to interpret. If the authors want to explore the relationship of Marital status, Household register, Urban and rural classification with the “Nations problems perceptions”, they might consider using other statistical procedures as ANOVA….

In Table 3, authors should also inform about the effect size of the variables. It seems that although they find significant relations, the effect sizes are very small (maybe insignificant), so they must inform about it, and comment it latter on the Discussion.

The manuscript is interesting, has an applied dimension and includes a big sample, but I only would recommend publishing it, after a Major Revision.

Author Response

Response to Reviewer 2 Comments

Dear Ms. Erica Wang and the reviewer,
Thank you for your constructive comments on our paper (ijerph-699193). We are very grateful to the editor and reviewers for the excellent level of detailed feedback offered to enable us to enhance the manuscript. We have carefully addressed the comments of the reviewers and highlighted(in red)the main changes made in the revised paper. Thank you for the opportunity to resubmit our manuscript for further consideration for publication in International Journal of Environmental Research and Public Health. All responses are made as follows.

Sincerely

Comments to the Author

Point 1: The authors present a national, large scale study to examine the potential mediating of community identity on the relationship between socioeconomic status and “sense of gain”. In my opinion, here we have the main problem of this research. As the authors will recognize in their limitations paragraph (line 219), the items that they used for the concept “sense of gain” do not reflect the meaning of the concept, so they should consider changing to another concept. For instance, it would be more accurate to speak about “National problems perceptions” than “sense of gain”.
Response 1: Thank you for your constructive advice for improving our study. First, like many previous studies, Wen and Liu[3] measured sense of gain through gap between the rich and the poor, government corruption and basic public services such as education, pension, medical treatment, housing, employment and social security. Other researchers[4] considered that sense of gain should be composed of economic sense of gain, public service sense of gain, political sense of gain, security sense of gain and self-realization sense of gain. Therefore, we still used the concept “sense of gain” in the full text. Second, we have added the reason we used these items for the concept “sense of gain” (line 138). Third, due to the limitations of a national, large-scale scale study, we further pointed out the future directions in the limitation(from line 242 to line 245).
“With reference to existing studies[3, 4], sense of gain was measured by asking the questions “How serious do you think the following problems are in our country” in this study, the following questions include environmental conservation, gap between the rich and the poor, employment, education, medical treatment, housing, social security, and government corruption.”
“First, data from 2016 China Family Panel Studies is a second-hand data. This may lead to the items selected to measure the variables cannot fully reflect the meaning of the concept. Specific research tools for sense of gain should be developed in future studies.”

References

  1. Wen, H.; Liu, Z. P. Time comparison of the sense of gain to China: Trends and disparities empirical analysis based on Chinese urban and rural social governance data. J Soc Sci. 2018, (3), 3–20, doi:10.13644/j.cnki.cn31-1112.2018.03.001.
  2. Yang, J. L.; Zhang, S. H. Analysis of the General Social Survey Data on the Chinese people's sense of fulfillment. J M Stud. 2019, 188(3), 102–112.

Point 2: In the Introduction (line 36) the definition they offer from the “spiritual level” of the concept “sense of gain” it is politically biased. They back on three references, which are far from being the most accepted. Again, I would recommend not using the concept “sense of gain”; but if the authors still want to mention it, do not include that definition of the spiritual level (they might consider other definitions included in the “Handbook of the Psychology of Religion and Spirituality”).
Response 2: Thank you for your constructive advice for improving our study. First, we removed sentences unrelated to the purpose of the study. Second, we have revised the definition of sense of gain to make it the most accepted(from line 34 to line 39). Third, we have removed the definition of the spiritual level.
“Sense of gain, as a Chinese concept proposed in recent years, has attracted more and more researchers' attention. Sense of gain refers to the improvement of material living standards brought by national development, including not only housing, education, medical care and social security, but also the right to enjoy fairness and justice[1]. Some studies[2] also believe that sense of gain is an overall feelings about the benefits of economic and social development, the realization of social equity and justice, and the supply of basic public services.”

References

  1. People's Daily Online. How to enhance the sense of happiness and security (three questions for people's livelihood for people's better life⑧). Available online: http://society.people.com.cn/n1/2017/1109/c1008-29635156.html (accessed on 9 November 2017)
  2. Li, P.; Bai, W. C. Research on the influence of people's sense of gain on government trust. Admin Tri. 2019, (4), 75–81, doi:10.16637/j.cnki.23-1360/d.2019.04.010.

Point 3: I believe authors do the right decision when they introduce some of the classical theories on social identity (line 55).
Response 3: We appreciate the reviewer raising this concern.

Point 4: In the methods, I would recommend that the authors arrange all the scores to be in the same direction, and the higher scores always represent higher values (Socioeconomic status, Community identity, Problems perceptions…).
Response 4: Thank you for your constructive advice for improving our study. First, we have arranged all the scores to be in the same direction(from line 134 to line 135 and from line 143 to line 144). Second, we have revised the relevant expression (from line 166 to line 201).
“We arranged all the scores to be in the same direction, and higher scores indicated higher community identity.”
“In order to better reflect the meaning of this concept, we conducted the reverse scoring, and higher scores indicated higher sense of gain.”
“3.1. Preliminary analyses
ANOVA was conducted to explore whether sense of gain differed between male and female, between agriculture account and non-agricultural account, between rural and urban, and between marital status. Results showed that there was no significant difference in sense of gain between male and female(F(1, 28299)=1.34, p>0.05). Significant difference was found for household register, with agricultural people being significantly more sense of gain than non-agricultural people(F(1, 28299)=287.88, p<0.01, and= 0.01). Sense of gain was significantly different between rural and urban, namely, rural individuals had more sense of gain than urban individuals(F(1, 28299)=287.884, p<0.01, and= 0.01). The significant difference was found for marital status, with widowed being significantly more sense of gain than others (F(4, 28295)=133.94, p<0.01, and=0.02).
As indicated in Table 2, results showed that age(r=0.30, p<0.01) was positively associated with sense of gain. Socioeconomic status was positively associated with community identity(r=0.20, p<0.01) and sense of gain(r=0.11, p<0.01). And community identity was positively associated with sense of gain(r=0.17, p<0.01).
Table 2. Correlations among central study variables.

M±SD
1
2
3
1. Age
46.50±15.75

2. Socioeconomic status
0.00±1.70
0.10**

3. Community identity
22.19±3.22
0.15**
0.20**

4. Sense of gain
30.33±13.88
0.30**
0.11**
0.17**
Notes: **p<0.01 (two-tailed); *p<0.05 (two-tailed).
3.2. Mediating effects of community identity
To test the mediation effect, we conducted a mediation analysis using bootstrapping procedures proposed by Preacher and Hayes with the PROCESS macro in SPSS[40]. Table 3 and Figure2 displayed the bootstrap result of the mediating effect of community identity between socioeconomic status and sense of gain.
As shown in Table 3 and Figure 2A, socioeconomic status was associated with sense of gain(c, β=0.07, 95%[0.059, 0.073]). As shown in Table 3 and Figure 2B, the socioeconomic status -community identity path(a, β=0.11, 95%[0.108, 0.121]) and the community identity-sense of gain path(b, β=0.16, 95%[0.145, 0.168]) were significant. The socioeconomic status-sense of gain path(c’, β=0.05, 95%[0.041, 0.055]) was also significant. The effect size of the variable was 0.01, 95%[0.005, 0.007]. In sum, results supported our hypothesis that socioeconomic status was significantly associated with sense of gain, and that community identity mediated the relationship of socioeconomic status and sense of gain.
Table 3. Test of mediation effects of community identity on the relationship of socioeconomic status to sense of gain: Bootstrap results.
Path/effect
Standardized
β
BootSE
BootLLCI
BootULCI
c(total effect)
0.07
0.01
0.059
0.073
a Socioeconomic status→ Community identity
0.11
0.01
0.108
0.121
b Community identity→ Sense of gain
0.16
0.01
0.145
0.168
c’ Socioeconomic status →Sense of gain
0.05
0.01
0.041
0.055
a×b(indirect effect)
0.02
0.01
0.016
0.020
R-sq_med
0.01
0.01
0.005
0.007
Note. Bias corrected and accelerated 95% CI, bootstrap resamples= 5,000. The 95% CI for the standardized result was produced with the bias corrected and accelerated option in the bootstrap dialogue box.
Figure 2. Tests of the theorized mediation model. (A) The total effect (socioeconomic status predicting sense of gain). (B) The indirect effect (with community identity as mediator). Indirect effect, a×b(β)=0.02, p<0.01. Standardized regression coefficients are displayed.  *p<0.05,  **p<0.01.”

References

  1. Preacher, K. J.; Hayes, A. F. Asymptotic and resampling strategies for assessing and comparing indirect effects in multiple mediator models. Behav Res Methods. 2008, 40, 879–891, doi: 10.3758/brm.40.3.879.

Point 5: In addition, the authors should inform about the reliability (Cronbach´s Alfa scores) of their composed variables (Socioeconomic status, Community identity, Problems perceptions…).
Response 5: Thank you for your constructive advice for improving our study. We have added the sentences to inform about the reliability of major variables(from line 135 to line 136 and from line 144 to line 145).
“The Cronbach’s alpha of the scale in this study was 0.68.
The Cronbach’s alpha of the scale in this study was 0.84.”

Point 6: In their results, there is an important error. We cannot use correlations for categorical variables (line 148), it is very complicated to interpret. If the authors want to explore the relationship of Marital status, Household register, Urban and rural classification with the “Nations problems perceptions”, they might consider using other statistical procedures as ANOVA….
Response 6: Thank you for your constructive advice for improving our study. We have used ANOVA to explore the relationship of categorical variables with sense of gain(from line 166 to line 174).
“ANOVA was conducted to explore whether sense of gain differed between male and female, between agriculture account and non-agricultural account, between rural and urban, and between marital status. Results showed that there was no significant difference in sense of gain between male and female(F(1, 28299)=1.34, p>0.05). Significant difference was found for household register, with agricultural people being significantly more sense of gain than non-agricultural people(F(1, 28299)=287.88, p<0.01, and= 0.01). Sense of gain was significantly different between rural and urban, namely, rural individuals had more sense of gain than urban individuals(F(1, 28299)=287.884, p<0.01, and= 0.01). The significant difference was found for marital status, with widowed being significantly more sense of gain than others (F(4, 28295)=133.94, p<0.01, and=0.02).”

Point 7: In Table 3, authors should also inform about the effect size of the variables. It seems that although they find significant relations, the effect sizes are very small (maybe insignificant), so they must inform about it, and comment it latter on the Discussion.
Response 7: Thank you for your constructive advice for improving our study. First, we have added the effect size of the variables(from line 186 to line 197). Second, we have added comment it latter on the Dicussion(from line 247 to line 250).
“As shown in Table 3 and Figure 2A, socioeconomic status was associated with sense of gain(c, β=0.07, 95%[0.059, 0.073]). As shown in Table 3 and Figure 2B, the socioeconomic status-community identity path(a, β=0.11, 95%[0.108, 0.121]) and the community identity-sense of gain path(b, β=0.16, 95%[0.145, 0.168]) were significant. The socioeconomic status-sense of gain path(c’, β=0.05, 95%[0.041, 0.055]) was also significant. The effect size of the variable was 0.01, 95%[0.005, 0.007]. In sum, results supported our hypothesis that socioeconomic status was significantly associated with sense of gain, and that community identity mediated the relationship of socioeconomic status and sense of gain.”
Table 3. Test of mediation effects of community identity on the relationship of socioeconomic status to sense of gain: Bootstrap results.
Path/effect
Standardized
β
BootSE
BootLLCI
BootULCI
c(total effect)
0.07
0.01
0.059
0.073
a Socioeconomic status→ Community identity
0.11
0.01
0.108
0.121
b Community identity→ Sense of gain
0.16
0.01
0.145
0.168
c’ Socioeconomic status →Sense of gain
0.05
0.01
0.041
0.055
a×b(indirect effect)
0.02
0.01
0.016
0.020
R-sq_med
0.01
0.01
0.005
0.007

“Third, although we find significant relationship between socioeconomic status, community identity and sense of gain, the effect size is very small. Future research should focus on other variables that affect sense of gain and the underlying mechanisms between them.”

The manuscript is interesting, has an applied dimension and includes a big sample, but I only would recommend publishing it, after a Major Revision.

Round 2

Reviewer 1 Report

Authors’ revisions addressed issues/concerns with original manuscript. A brief explanation or should be added to discussion for two key and apparentely unexpected findings: 1) “Sense of gain was significantly different between rural and urban, namely, rural individuals had more sense of gain than urban individuals”; and 3) “The significant difference was found for marital status, with 174 widowed being significantly more sense of gain than others”.

Author Response

Response to Reviewer 1 Comments

Dear Ms. Erica Wang and the reviewer,
Thank you for your constructive comments on our paper (ijerph-699193). We are very grateful to the editor and reviewers for the excellent level of detailed feedback offered to enable us to enhance the manuscript. We have carefully addressed the comments of the reviewers and highlighted(in blue)the main changes made in the revised paper. Thank you for the opportunity to resubmit our manuscript for further consideration for publication in International Journal of Environmental Research and Public Health. All responses are made as follows.

Sincerely

Comments to the Author

Point 1: Authors’ revisions addressed issues/concerns with original manuscript. A brief explanation or should be added to discussion for two key and apparently unexpected findings: 1) “Sense of gain was significantly different between rural and urban, namely, rural individuals had more sense of gain than urban individuals”; and 3) “The significant difference was found for marital status, with 174 widowed being significantly more sense of gain than others”.
Response 1: We appreciate these suggestions. We have added to discussion for two key and apparently unexpected findings(from line 209 to line 217).
“An apparently unexpected finding in the study was that rural individuals had more sense of gain than urban individuals. In recent years, China has implemented the strategy of rural revitalization, with continuous economic development in rural areas and gradual improvement in the social security system. Sense of gain can also be the result of vertical social comparisons, where current conditions are compared to historical conditions. As a result, individuals in rural areas may have higher sense of gain than urban individuals[4, 12]. And another unexpected finding was that widowed being had significantly more sense of gain than others. Widowed beings get more help from society. And because widowed beings tend to be older, they are more likely to experience the positive effect of China’s rapid economic growth[12], which enhance their sense of gain.”

References

  1. Yang, J. L.; Zhang, S. H. Analysis of the General Social Survey Data on the Chinese people's sense of fulfillment. J M Stud. 2019, 188(3), 102–112.
  2. Wang, T.; Tan, Y. F.; Fu, X. S. The residents' sense of gain in China and its determinants. Financ Econ. 2018, (9), 120–132.

Reviewer 2 Report

The authors hace considered all the recommendations, and have improved their work. 

Author Response

Response to Reviewer 2 Comments

Dear Ms. Erica Wang and the reviewer,
Thank you for your constructive comments on our paper (ijerph-699193). We are very grateful to the editor and reviewers for the excellent level of detailed feedback offered to enable us to enhance the manuscript. We have carefully addressed the comments of the reviewers and highlighted(in blue)the main changes made in the revised paper. Thank you for the opportunity to resubmit our manuscript for further consideration for publication in International Journal of Environmental Research and Public Health. All responses are made as follows.

Sincerely

Comments to the Author

Point 1: The authors have considered all the recommendations, and have improved their work.
Response 1: Thank you for your constructive advice for improving our study.
